# Egg-sac-brooding wolf spiders show flexible hatchling emergence and context-dependent escape performance

**Bai-Lu Chen**[1,2,*]**, Jing-Xin Liu**[1,*] **and Zhanqi Chen**[1,‡]

## ABSTRACT

Egg-sac brooding is a costly maternal strategy for which evolutionary persistence hinges on clear offspring benefits and effective maternal tactics to offset those costs. Using the wolf spider *Pardosa pusiola*, we examined (1) whether hatchling emergence depends on the presence of a conspecific mother, (2) whether egg sac opening is a flexible response to embryonic cues, and (3) how mothers modulate locomotor performance under different ecological risks (sun exposure, flooding, predation). Conspecific foster mothers matched biological mothers in synchronizing egg-sac opening with embryonic development, whereas interspecific foster mothers (*Pardosa astrigera*) mistimed opening in most cases. Motherless egg sacs contained fully developed but un-emerged hatchlings, confirming that maternal presence is indispensable for emergence, not for hatching itself. Under moderate sun exposure, egg-sac-carrying females escaped slower than non-carrying females. Under high sun exposure or predator stimulus, carrying females escaped as fast as or faster than non-carrying females. Under simulated flooding, carrying females suffered higher mortality, yet survivors showed no difference in escape speed compared to non-carrying females. These results demonstrate flexible egg-sac management coupled with adaptive maternal locomotion, illustrating how costly parental care can be maintained when parents adjust their behavior according to environmental risk.

KEY WORDS: Egg-sac brooding, Embryo–mother communication, Parental care, Risk avoidance, Spiderling emergence

## INTRODUCTION

Parental care is the combination of parental traits that benefit the survival and growth of offspring, often at the cost of the parents' own survival and future reproduction (Clutton-Brock, 1991; Omkar, 2022; Royle et al., 2012). Parental care plays a central role in the evolution of parental traits, and extremely diverse forms of parental care have been reported (Omkar, 2022; Royle et al., 2012), such as gestation, provisioning, and protection from predators and environmental stresses. After laying eggs or giving birth to the juveniles, many animals, such as the majority of fishes, amphibians, and arthropods, stop parental care. Some species, such as some amphibians, fishes, and herbivorous bugs, continue to stay with eggs/juveniles for a short period to protect them from environmental hazards and predators, and a small proportion of species perform egg brooding, where parents (usually mothers) carry eggs externally or internally after laying and provide various care during carrying (Baur, 1994; Machado and Trumbo, 2018; Omkar, 2022; Royle et al., 2012). Although there are reports of egg brooding in vertebrates, such as the well-known mouth brooding behavior in some fishes and amphibians (Luiz et al., 2024; Ringler et al., 2023; Royle et al., 2012), egg brooding is more common and diverse in invertebrates (Royle et al., 2012), especially arthropods (Machado and Trumbo, 2018; Royle et al., 2012), such as in hermit crabs *Pagurus bernhardus* (Neil and Elwood, 1985), mosquitoes *Trichoprosopon digitatum* (Lounibos and Machado Allison, 1983), and wolf spiders (family Lycosidae), fish-eating spiders (Pisauridae), spitting spiders (Scytodidae), and fishing spiders (Trechaleidae) (Ewunkem and Agee, 2022; Foelix, 2011; Jocqué and Dippenaar-Schoeman, 2007).

Egg brooding may improve eggs' survival and hatching rate as brooders might: (1) protect eggs from predators, parasitoids, pathogens and harsh biotic stresses; (2) actively regulate the microenvironment, such as the oxygen, humidity, light, and temperature, to create suitable conditions for embryonic development (Abecia et al., 2022; Humphreys, 1974; Li, 2002; Omkar, 2022; Ringler et al., 2023; Royle et al., 2012); or (3), as reported in several wolf spiders, the mothers might be able to sense the embryonic development and expand the egg sac accordingly, and then open the sac to help the hatchlings emerge (Eason, 1964; Fujii, 1978; Ruhland et al., 2019; Ruhland et al., 2017). Thus, it is reasonable to posit that eggs benefit from a brooding mother who can respond to their developmental needs. However, empirical studies indicate that egg brooding is costly and often has a negative impact on brooder performance as brooding activities (1) are time-consuming and energy-demanding, as carrying eggs often decreases the brooder's foraging efficiency (Colancecco and Rypstra, 2007; Ruhland et al., 2016a,b), and (2), it increases the risk of brooders dying, as egg-carriers might be more conspicuous and/or slower to escape from predators and extreme environmental risks such as heat and drought (Argaez and Munguía-Steyer, 2023; Clutton-Brock, 1991; Colancecco and Rypstra, 2007; Royle et al., 2012; Suzuki and Futami, 2018).

While the fitness trade-offs of egg brooding have been extensively quantified across diverse taxa, three unresolved questions are central to understanding this system. (1) Is egg sac expansion and opening a fixed behavior with little or no plasticity, or a flexible response to embryonic cues? (2) Are mother–embryo cues for egg-sac opening species-specific? (3) How do brooding females adjust their escape performance from different threats? We address these questions by combining controlled brooding experiments with survival-based escape assays under ecologically relevant risks. Based on these questions, we propose the following hypotheses: (1) egg-sac expansion is a behaviorally flexible, rather than an inflexible, process; (2) the cue

[1]Yunnan Key Laboratory of Forest Ecosystem Stability and Global Change, Xishuangbanna Tropical Botanical Garden, Chinese Academy of Sciences, Yunnan 666303, China. [2]University of Chinese Academy of Sciences, Beijing 100049, China.
*These authors contributed equally to this work

‡Author for correspondence (chenzhanqi@xtbg.ac.cn)

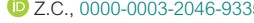 Z.C., 0000-0003-2046-9335

triggering opening is species-specific, heterospecific fosters will mistime opening relative to conspecific fosters; (3) successful emergence of hatchlings depends on maternal presence, independent of hatching per se; (4) carrying an egg sac incurs measurable costs (slower speed or higher mortality under benign conditions); (5) when risk is acute, brooding females offset the costs by increasing escape speed.

Free-moving wolf spiders (Araneae: Lycosidae) are excellent candidates to explore these questions as this family encompasses 2489 species (WCA, 2025) and are well-known for their continuous brooding behavior. After laying eggs, females attach the egg sac to their spinnerets and carry it around wherever they go for 2-4 weeks continuously until the eggs are hatched, then they open the sac to help the hatchlings emerge (Fig. 1). They then continue to brood the hatchlings on their back for about 1 week until the dispersal of the spiderlings (Foelix, 2011; Jocqué and Dippenaar-Schoeman, 2007). Thus, we selected the wolf spider *Pardosa pusiola* (Thorell, 1891) as study species in this work. *P. pusiola* is widely distributed from India, Sri Lanka, and Southeast Asia to southern China (GBIF Secretariat, 2023; Zhang and Wang, 2017). Besides natural forests edges, shrub lands and grasslands, they are also common in man-made ecosystems, such as gardens, agricultural fields, rubber plantations, tea plantations, etc. (Li et al., 2017; Nasir et al., 2016; Omelko and Marusik, 2020; Wang and Zhang, 2014; Zhang and Wang, 2017). The brooding behavior of *P. pusiola* has not been studied before, but our preliminary observation indicates that it exhibits similar brooding behavior to other congeneric species, such as *Pardosa astrigera* and *Pardosa milvina* (Berry and Rypstra, 2021; Fujii, 1978). In this study, we first tested egg-sac expansion flexibility and brooding-behavior specificity, then compared the escape-behavior differences between egg-brooding and egg-removed females under sun-exposure (Fig. 2A) and 'flood' (Fig. 2B) conditions.

## RESULTS

On average, the weight of the egg sac was 86% of the mother's post-reproductive body weight (female body weight: 0.026±0.001 g, $N=40$; egg sac weight: 0.022±0.001 g, $N=40$).

### Egg sac expansion flexibility and species specificity of mother brooding behavior

All 38 egg sacs brooded by biological or conspecific foster mothers were opened on time, whereas, although none of the 28 un-brooded egg sacs were opened by the hatchlings from inside, they all contained dead hatchlings when we hand-opened them. This result confirmed that brooding process might not be vital for the embryos' development and hatching, but hatchlings depend on brooders to expand and open the sac to emerge.

All 26 egg sacs brooded by biological mothers and all 12 egg sacs brooded by conspecific foster mothers were opened on time (Fisher's exact test: $P=1$), and their eggs showed similar hatching probabilities [biological mother: 0.96±0.12 (mean±s.d.); conspecific foster: 0.97±0.08; Bayesian logistic mixed model (BLMM): $\beta_{BM}-\beta_{CSF}=0.98$, 95% CI-HPD=(−3.50, 5.05)] (Fig. 3). Given that the conspecific foster-brooded egg sacs were unlikely

to be at the same developmental stages compared with their original sacs, this result suggests that the biological mothers and conspecific fosters performed equally well in brooding egg sacs, which demonstrates that *P. pusiola* mothers' brooding behavior is flexible, and that they expand egg sacs according to the developmental stage of the egg sacs they brooded.

Of the 24 egg sacs brooded by interspecific foster mothers, 18 (75%) were opened, four (17%) were abandoned and two (8%) were eaten (Fig. 4). Of the opened egg sacs, four (22%) were opened on time (Fig. 5A,B), eight (45%) were opened late (Fig. 5C,D), and six (33%) were opened early (Fig. 5E,F). Although the ratios of opened egg sacs were not statistically different between conspecific foster mothers and interspecific foster mothers ($P=0.159$), interspecific foster mothers opened a significantly lower proportion of egg sacs on time than conspecific foster mothers ($P\ll0.001$). Conversely, the hatching possibilities of eggs in sacs brooded by interspecific fosters were similar to those in egg sacs brooded by conspecific fosters [interspecific foster: 0.93±0.13; $\beta_{CSF}-\beta_{ISF}=1.50$, 95% CI-HPD=(−2.86, 5.70)] (Fig. 3). These results, together with the fact that un-brooded egg sacs all had hatched eggs, but hatchlings could not open the sac to emerge, demonstrate that in *P. pusiola* egg sacs may not need brooding, and that the mother is able to adjust egg-sac openings with developmental stage.

### Mothers escaping from risk
#### Sun exposure

Under a medium level of sun exposure, all 25 egg-sac-carrying and 25 egg-sac-removed *P. pusiola* females successfully reached the shelter, with their escaping speeds 2.0±1.0 cm s⁻¹ and 7±8 cm s⁻¹, respectively. Under a high level of sun exposure, 95.92% (47/49) of the egg-sac-carrying and 95.83% (46/48) of the egg-sac-removed *P. pusiola* females successfully reached the shelter with speeds of 10.5±4.0 cm s⁻¹ and 9.5±4.2 cm s⁻¹, respectively.

There was strong statistical evidence that under medium sun exposure, egg-sac-removed females escaped much faster than egg-sac-carrying females [$\beta_{No\_sac}-\beta_{With\_sac}=4.77$, 95% CI-HPD=(1.38, 8.22)], under a high level of sun exposure, egg-sac-removed females' escape speed did not significantly differ from egg-sac-carrying females [$\beta_{No\_sac}-\beta_{With\_sac}=-1.08$, 95% CI-HPD=(−2.69, 0.73)] (Fig. 6A,B). Statistical analysis also showed that, when exposed to strong sun, egg-sac-carrying females escaped at a much faster speed compared to under medium sun exposure [$\beta_{H\_sun}-\beta_{M\_sun}=8.77$, 95% CI-HPD=(7.52, 10.10)], while egg-sac-removed females did not show significantly different escape speed under different levels of sun exposure ($\beta_{H\_sun}-\beta_{M\_sun}=2.97$, 95% CI-HPD=(−0.70, 6.52)] (Fig. 6A,B).

#### Flood

In the water tank, 85.7% (24/28) egg-sac-removed *P. pusiola* females and 50% (18/36) egg-sac-carrying females successfully reached the island (safety site), and the escape speeds were 9.7±6.1 cm s⁻¹ and 9.0±4.9 cm s⁻¹, respectively. Although there was strong statistical evidence that carrying egg sacs significantly reduced the success rate of escaping from the 'flood' scenario (Fisher's exact tests: $P=0.004$),

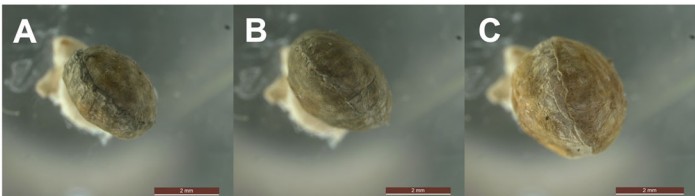

**Fig. 1. Wolf spider egg sacs at successive developmental stages.** (A) Freshly laid (1–2 days): smallest, silk tightly packed. (B) Mid-stage (~7 days): visibly distended, silk loosened, increased translucency. (C) Late-stage (~14 day): maximal volume, spiderling outlines detectable, maternal silk tension minimized. The gradual expansion is an actively moderated, loss-by-relaxation process controlled by the brooding female. Photos by B.-L.C.

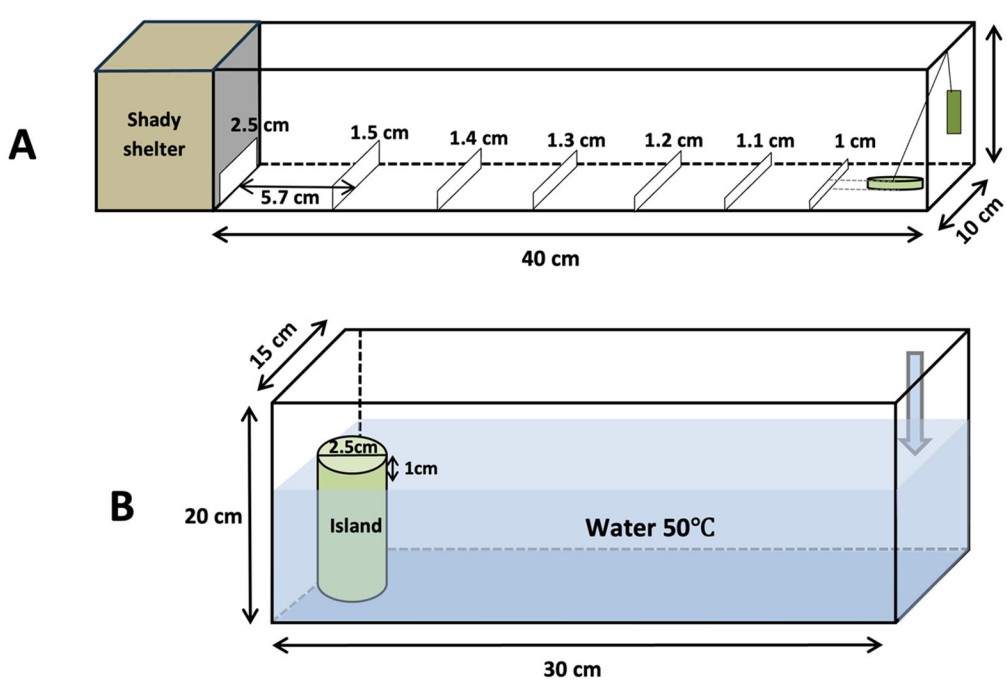

**Fig. 2. Apparatus for sun exposure and 'flood' escape tests.** (A) Spiders were acclimated in the green plastic cup (diameter: 1.5 cm, height: 1 cm) under a cover for 1 min before the test. The cover was then removed by pulling the string to release the spider and begin the trial. The corridor contained seven obstacles with heights increasing from 1 cm at the first obstacle to 1.5 cm at the sixth obstacle in increments of 0.1 cm, and to 2.5 cm at the seventh obstacle. The other end covered by brown paper is the shaded sheltering space for spiders to avoid direct sun exposure. (B) The water tank of a thermostat water bath (DRHH-2). Spiders were released into the water (arrow point) to start the test, and the plastic 'island' (diameter: 2.5 cm, height above the water: 1 cm) at the other end of the water tank was the safety site.

the escape speeds of successful spiders in the two groups were not different [$\beta_{No\_sac}-\beta_{With\_sac}$=0.72, 95% CI-HPD=(−2.80, 4.17)] (Fig. 6C,D). We recorded whether females dropped their egg sacs during escape trials and abandonment occurred in only three out of 36 flood tests (8%), all within the last 5 s before submersion.

### Predator

In a simulated-predator trial using a hand-held shaker, all 25 egg-carrying and all 25 egg-sac-removed *P. pusiola* females successfully reached the shelter. The escape speeds of egg-sac-removed spiders (6.6±4.0 cm s$^{-1}$) were significantly slower than that of egg-sac-carrying spiders (9.3±5.0 cm s$^{-1}$) [$\beta_{No\_sac}-\beta_{With\_sac}$=−2.76, 95% CI-HPD=(−5, −0.22)] (Fig. 6E,F).

### DISCUSSION

The present study delves into the maternal strategy of egg-sac carrying in wolf spiders, shedding light on its significance for hatchling emergence, and adaptive maternal behaviors under different risk conditions. Our findings reveal three critical dimensions of this maternal strategy: (1) egg-sac expansion is a flexible process based on

the developmental stages of the embryos; (2) mother–embryo communication exhibits species-specific cues for egg sac expansion and hatchling emergence; (3) and the context-dependent locomotor adaptations that prioritize successful emergence of the current clutch over the mother's own survival under extreme threats. These results reveal how risk-induced shifts in female escape performance (faster sprinting under predation or heat stress) feed back into the fine-tuning of brooding behavior – namely the timing of egg sac opening and the reluctance to abandon the clutch. We use the term 'cues' throughout the Discussion with the understanding that we cannot distinguish between cues and signals in the strictest sense. Our results show that mothers responded to developmental cues from embryos, but we cannot confirm that these cues have been selected for communication.

The first experiment revealed that the mother's presence is crucial for hatchling emergence in *P. pusiola*. Whether the egg sac was brooded by its biological mother or a conspecific foster, all brooded egg sacs were opened on time, enabling the hatchlings to emerge. However, none of the un-brooded egg sacs were opened by hatchlings from within, and all contained dead hatchlings and

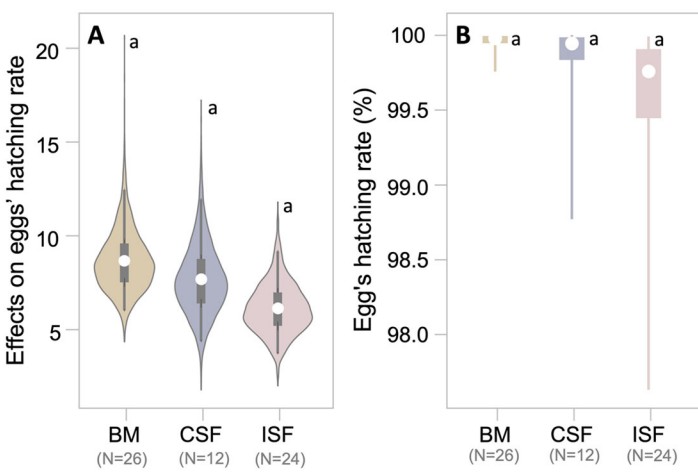

**Fig. 3. Eggs of *P. pusiola* brooded by biological mothers (BM), conspecific fosters (CSF) and interspecific fosters (ISF) have a similar hatching possibility.** (A) Posterior distributions of the effects of BM (beige), CSF (light blue) and ISF (pink beige) on hatching success of *P. pusiola* eggs. Violins indicate densities, thin and wide rectangles represent 50% and 95% credible intervals, respectively, and white points indicate posterior mean estimates. Note that the effects are in logit space, and thus indicate an effect on egg log-odds ratio of successful hatching. (B) Posterior conditional effects of BM, CSF and ISF on the egg-hatching probability. Points (white) indicate the means of predicted probabilities of egg brooded by different types of brooders. Error bars indicate 50% and 95% credible intervals as in A. Statistical note: Letters (a,b) above violins indicate credible groupings based on BLMM posterior distributions. Groups sharing the same letter do not differ credibly (95% CI-HPD includes zero). Sample sizes (N) for each group are indicated below the group labels.

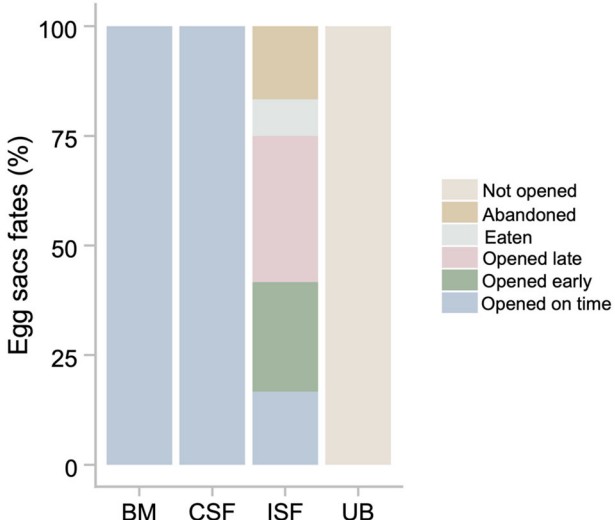

**Fig. 4. Fates of egg sacs brooded by biological mothers (BM), conspecific fosters (CSF), interspecific fosters (ISF), and unbrooded (UB).**

unhatched eggs when manually opened by the experimenter. This finding underscores that while maternal care may not be essential for egg hatching, it is indispensable for hatchling emergence. It appears that the hatchlings cannot break free from the egg sac without the mother's assistance in expanding and opening it at the right time,

which is consistent with findings in other egg-brooding spiders like *Pardosa milvina* (Berry and Rypstra, 2021) and *Pardosa lapidicina* (Eason, 1964). The biological mothers and conspecific fosters performed equally well in brooding the egg sacs, regardless of whether the developmental stage of the accepted egg sac matched their own. This implies that the brooding behavior of *P. pusiola* females is flexible and that they can adjust their brooding behavior, such as with egg sac expansion, based on cues from the developing embryos. However, when the egg sacs were brooded by interspecific fosters (*P. astrigera* females), the results were less favorable. Although *P. astrigera* females did open the egg sacs, the timing was often incorrect. This indicates that the communication cues between hatchlings and brooders in wolf spiders are species-specific.

The second experiment compared the escaping speeds of egg-sac-carrying females versus egg-sac-removed females under different stress conditions, namely sun exposure, 'flood', and 'predator'. The results showed that under medium-level sun exposure, egg-sac-removed females escaped much faster than egg-sac-carrying females. However, under high-level sun exposure, the escaping speeds of the two groups were comparable. In the flood experiment, fewer egg-sac-carrying females successfully reached the safety island, but those that did escape had comparable speeds to egg-sac-removed females. In the predator-simulated experiment, egg-sac-carrying females demonstrated a significant advantage over egg-sac-removed females in their escape speed. These findings

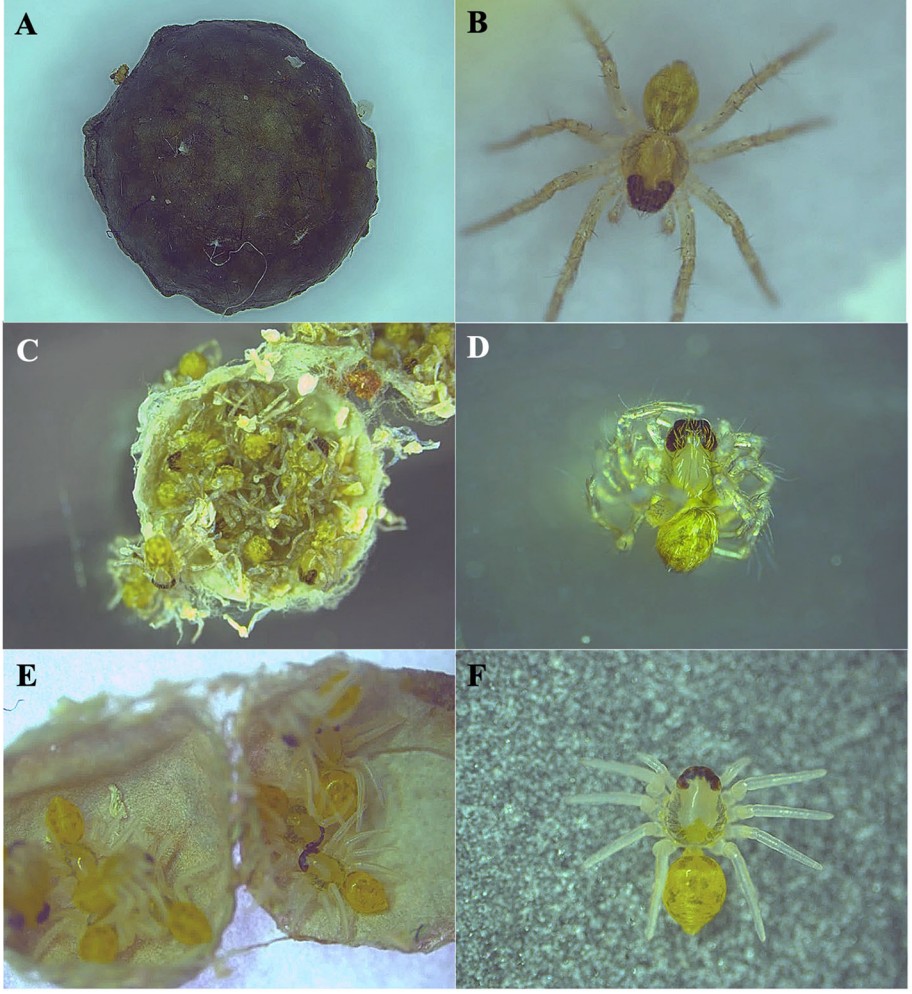

**Fig. 5. Fates of egg sacs and status of hatchlings under different brooding condition**. (A) An unopened egg sac in the end of the experiment from the un-brooded group (UB). (B) A healthy hatchling emerged from egg sacs brooded and opened on-time by biological mothers (BM) or conspecific fosters (CSF). (C,D) A late-opened egg sac with full-developed dead hatchlings from the interspecific foster group (ISF). (E,F) An early-opened egg sac with underdeveloped, non-viable hatchlings from the ISF.

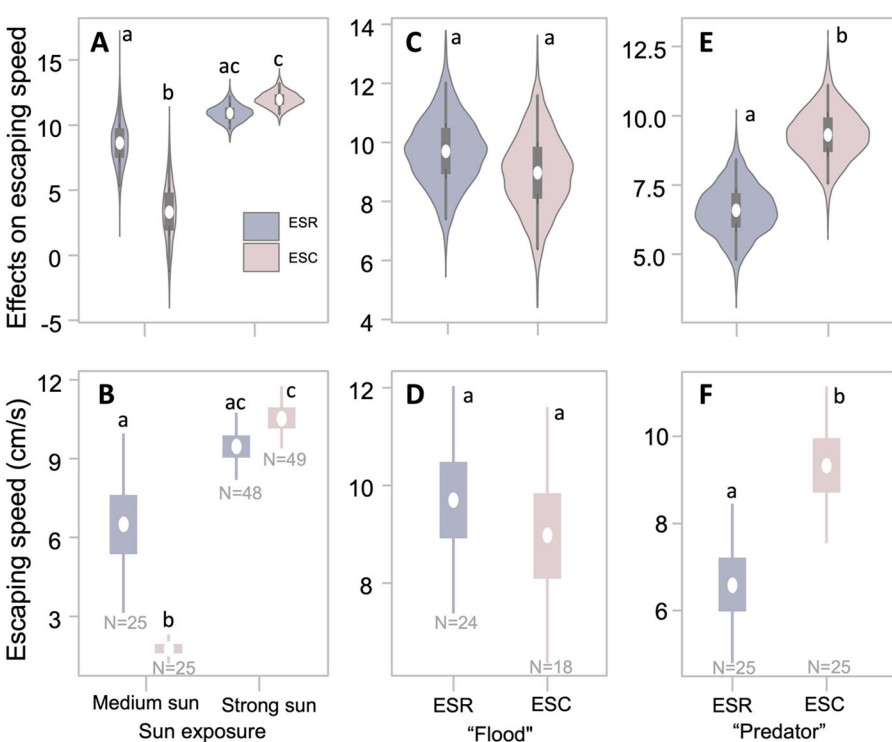

**Fig. 6. Escape performances of egg-sac-carrying (ESC) and -removed (ESR) *P. pusiola* mothers under different risks.** (A,B) Sun exposure: posterior distributions (A) and conditional effects (B) of escape speed under medium (M_sun) and high (H_sun) levels of sun exposure. (C,D) Simulated flooding: posterior distributions (C) and conditional effects (D) of escape speed for individuals that successfully reached safety. (E,F) Simulated predator: posterior distributions (E) and conditional effects (F) of escape speed. In panels A, C, and E, violins indicate posterior densities, thin and wide rectangles represent 50% and 95% credible intervals, respectively, and white points indicate posterior mean estimates. In panels B, D, and F, points indicate the means of predicted escape speeds, and error bars represent 50% and 95% credible intervals. BLMs were used to compare escape speeds. Letters (a,b) indicate credible groupings. Sample sizes (N) for each group are provided in the figure.

suggest that egg-sac-carrying females can modulate their escape behavior based on the context, showing enhanced performance under acute threats like predation and high heat, despite the inherent costs revealed in the flood scenario.

The most significant evidence was the result in the flood scenario. Compared to sun exposure and predator, flood was the most dangerous, as both egg-sac-carrying and -removed females had failed (dead) individuals in this test, thus it is reasonable to posit that every tested individual tried their best to escape to the safety island in this test. Therefore, although it was true that carrying an egg sac was extremely dangerous in the flood scenario (as more egg-sac-carrying *P. pusiola* females failed to reach the island), comparable escape speeds and the fact that no egg-sac-carrying females dropped their sacs before dying in the water or before reaching the island suggests that egg-sac-carrying females' escape capacity should be boosted, and that the females prioritized reproduction over their own survival. More evidence that brooding females boosted their risk-avoidance capacity was that egg-sac-carrying *P. pusiola* females escaped significantly faster than egg-sac-removed females in the simulated predator condition. It has been reported that egg-sac-carrying *P. milvina* females receive higher rates of predation compared to egg-sac-removed females (Colancecco and Rypstra, 2007). However, that experiment was conducted with a real predator and in a limited space for 2 h, which means that the *P. milvina* females had no place to escape to, while real threats in the field are usually in open areas and the predation risks typically last for just a few seconds. We provided a 40 cm escape passageway, and the results indicate that the egg-sac-carrying females were able to escape more efficiently than egg-sac-removed females facing a 'predator' even with the 'burden' of the egg sac. These suggest that outcome/costs of brooding depend on the environment and the severity of risk and probability of survival.

The boosted locomotor efficiency of egg-sac-carrying females facing 'flood' and 'predator' might be due to a temporary

over-consumption of energy, and a previous study on the energetic cost of egg sac brooding for mothers in *P. saltans* may support this hypothesis (Ruhland et al., 2016b). Alternatively, the similar or faster escape speeds of brooding females under high-risk conditions suggest context-dependent adjustment of locomotor performance rather than a simple energetic boost. Under moderate sun exposure, the slower escape of brooding females may reflect a trade-off between thermoregulation benefits and escape speed. As the escaping speed of egg-sac-carrying females was boosted under serious risks ('flood' and 'predator') and egg-sac carrying might not affect the locomotor performance under safe temperatures (Villaseñor-Amador et al., 2025), our result that under a medium-level of sun exposure egg-sac-carrying *P. pusiola* females reached the shelter at a much slower speed compared to egg-sac-removed females suggests that a medium-level of sun exposure is not a serious stress for brooding females, instead they may take the opportunity to sun bask their sacs (Humphreys, 1974). However, the high level of sun exposure might be a real risk as egg-sac-carrying *P. pusiola* females escaped significantly faster under a high level of sun exposure compared to under medium-level sun exposure.

The key evolutionary drivers for the faster escape behavior observed in egg-sac-carrying females are likely high predation risks and the vulnerability of spiderlings under extreme environmental conditions. The specialized spider-eating spider *Portia labiata*, for example, preferentially preys on egg-sac-carrying females of the subsocial spitting spider *Scytodes pallida* (Li and Jackson, 2003). Additionally, spiderlings are generally more fragile than adults, making the protection of egg sacs crucial for the survival of the offspring (Clutton-Brock, 1991; Foelix, 2011; Klug and Bonsall, 2010; Royle et al., 2012). These factors may have led to the evolution of enhanced locomotor efficiency in egg-sac-carrying females as a means to survive predation and extreme environmental conditions. In fact, the measured aggressive response was independent of clutch size and female body size (Nascimento and Gonzaga, 2015), indicating that females will invest their valuable

energy if their egg sacs are threatened. Therefore, egg-sac-brooding mothers experience elevated selection pressure that may promote adaptations in order to survive predation and/or extreme environmental conditions.

The higher mortality of brooding females under flooding conditions highlights a significant cost of brooding. First, the flood experiment indicated that egg-sac-carrying females are less likely to survive flooding events. This may be due to the egg sac becoming heavier when wet, thereby reducing the mother's mobility. Alternatively, the observed behavior of egg-sac-carrying mothers raising their abdomen to avoid water contact may also contribute to this disadvantage. Given the frequent flooding in their natural habitat, egg-sac-brooding may impose significant costs on this species (Ruhland et al., 2019). However, the similar escape speeds of survivors suggests that brooding females are capable of matching the speeds of non-brooding females when they do escape in flooding conditions.

In conclusion, our study comparatively tested the importance of the egg-sac-brooding maternal strategy in eggs' hatching, hatchling emergence and the mother's survival abilities under different risk conditions during egg-sac brooding. We found that the mother's presence is indispensable for the emergence of the young, and that egg-sac-carrying mothers escaped more efficiently under extreme risks compared to egg-sac-removed females. These results illustrate that the egg-sac-brooding maternal strategy may have evolved due to the reliance of hatchling development and emergence on the mother's ability to adjust egg sac tightness across different developmental stages. Additionally, mothers adopting the egg-sac-carrying behavior may have evolved enhanced locomotor capacity to escape from extreme risks, despite the potential costs associated with this strategy. Whether *P. pusiola* females ever strategically abandon their egg sacs remains an open question. Our pilot trials showed <10% spontaneous dropping under extreme flood risk, but a dedicated experiment – manipulating sac age, female nutritional state, and predator distance – could clarify the decision rule for 'sacrifice the clutch versus self-survival' (future work). We acknowledge several limitations of our study: the smaller sample size in the conspecific foster group, the lack of direct measurement of embryonic development stages, and the potential mismatch between our flooding simulation and natural flood conditions. Future studies could address these limitations by quantifying developmental stages more precisely and testing escape behavior under more naturalistic conditions.

## MATERIALS AND METHODS
### Spider collection and rearing
*P. pusiola* were collected from Xishuangbanna Tropical Botanic Garden, Chinese Academy of Sciences (XTBG, 21° 55′ N, 101° 16′ E), Yunnan, China, during the main breeding season (May to October) of 2016. The habitat was a shady mango orchard, with the ground covered by leaves and rotten fruit producing a food rich environment. *P. pusiola* are ground dwellers with limited climbing ability. Females produce relatively large egg sacs (weight: egg sac/female=86%) and brood eggs and carry new hatchlings for 3–4 weeks in total. The climate of this region experiences two well-defined seasons: a rainy season from May to October, and a dry season from November to April. Therefore, the females experience the challenges of a changing microenvironment, such as sun exposure and daily heavy rainfall, as well as predators, such as lizards and wasps, during the rainy season (main breeding season).

Once the study individuals were collected, they were transported to the laboratory in XTBG, kept in individual plastic cylindrical containers (diameter: 5 cm; height: 7 cm), and maintained in laboratory conditions. Each container had a piece of water-saturated sponge to maintain a constant relative humidity (70–80%). Laboratory temperature and light were left at ambient levels (matching the local tropical climate) to preserve the natural

thermal and photic conditions that females routinely encounter while brooding in the field; artificial regulation was avoided to maintain ecological realism. Individuals with missing legs were excluded from the experiment. For our experiment, females of *P. astrigera* L. Koch, 1878, an egg-sac-carrying congeneric spider with a similar body size from the same habitat in XTBG, were also captured for cross species egg-sac-brooding study.

The two species of spiders used in this study are not endangered or protected in China, thus we did not require special permit to collect them from the wild. The experimental design and data collection procedures were approved by the Ethics Committee of XTBG. All the living spiders were released back to their original collecting sites after the experiment was finished. All experiments used a between-subjects design. Each spider was tested only once under each condition.

### Experiment: egg-sac expansion flexibility
To determine whether egg-sac expansion is behaviorally flexible, we employed a 'cross-fostering' method for egg-sac brooding. Specifically, we designed three different fates for the egg sacs: group 1: biological mother brooded (BM): the egg sacs were detached and immediately returned to their biological mothers for reattachment and continued brooding (N=26). Group 2: conspecific foster brooded (CSF): the egg sacs (N=12) of *P. pusiola* were detached and immediately given to randomly chosen foster mothers (N=12) for continued brooding. Since the egg sacs were unlikely to have been produced on the same day (collected randomly from the field), the embryos' developmental stages could be earlier or later than those of the foster mothers' embryos. The ages of the egg sacs were not paired or determined in the egg-sac-swapped groups to confirm it is the internal nymphal cues at different developmental stages that induce the mother's egg sac tightness and opening manipulation behaviors. Thus, if the embryo hatching rate and the number of new hatchlings emerging from these egg sacs are similar to those in group 1, it suggests that female *P. pusiola* likely have the capacity to expand or open the egg sac flexibly based on the cues from the embryos inside the egg sac. Group 3: un-brooded (UB): the egg sacs were removed from their mothers and placed separately into individual containers (under the aforementioned temperature and humidity conditions) (N=28). This group was designed to test the importance of maternal presence in embryonic development and hatchling emergence. We monitored the fates of the three groups of egg sacs and recorded the following data: the hatch rate of embryos; whether the egg sacs opened or not without a mother presence; the number of spiderlings emerging from the egg sacs.

All individuals with egg sacs were collected on the same day. The egg sac swapping and mother removal experiments were conducted the next day. The individuals were then housed separately using the aforementioned method and environmental conditions until the egg sacs were opened, eaten, or abandoned. We collected fruit flies (*Drosophila melanogaster*) from the same habitat as the spiders and fed them to the spiders weekly starting from the day of collection. To detach an egg sac, we immobilized the female carrying the egg sac by trapping her with a soft brush, then grabbed the egg sac with forceps and pulled it away from the abdomen. After checking whether the egg sac was damaged, we placed it into a labelled, clean Petri dish. We then released the female back to her container and observed her for about 10 min to ensure she was not injured. After this period, uninjured egg sacs were placed into containers with different uninjured egg-sac-detached females (brooders) as described above (BM, CSF, or UB group). All brooders attached and began brooding the allocated egg sacs within 1 min.

### Experiment: species specificity of brooding behavior
To examine whether the spider egg-sac expansion and opening is species-specific, we designed an interspecific foster brooded group (ISF). We detached egg sacs of *P. pusiola* and gave them to females of *P. astrigera*, whose original egg sacs were detached, for brooding (N=24). We chose *P. astrigera* for the interspecific swap experiment because this species coexists with *P. pusiola* in the same environment, and they have similar body sizes, overlapping breeding seasons, and the same parental care strategies (egg brooding). The experimental process was identical to that of the conspecific foster brooded group described above.

## Experiment: mothers escaping from risks

To test our hypothesis that brooding mothers employ rapid escape strategies to mitigate risks through enhanced locomotory capacity, we compared the escaping speeds of two groups of *P. pusiola* mothers under challenging conditions. Group 1: egg-sac-carrying (ESC) mothers, whose egg sacs were removed and reattached following the procedure for BM mentioned above; and group 2: egg-sac-removed (ESR) mothers, whose egg sacs were removed but not reattached (we did not directly use the females not carrying an egg sac in the field in this experiment due to the difficulty of estimating whether they were newly matured females or females that had completed reproduction). We collected the spiders and conducted the egg sac removal and reattachment in the same day, then let them acclimate in laboratory for 1 day before the tests.

To challenge our spiders, we tested two common abiotic (sun exposure, and 'flood') and one biotic (predator) stresses. Sun exposure increases the temperature of the air and ground, and spiders have to find a shaded area to avoid overheating when the temperature is too high. As *P. pusiola* breeds in the rainy season, egg-sac-carrying females may experience sudden 'flood', which is dangerous if they cannot escape in time. They are also frequently predated by lizards, wasps, and other spiders.

### Sun exposure

We designed an apparatus to test the spiders' escape behavior under direct sun exposure in the morning between 10:00 and 11:00 (supposed to be the medium-level sun exposure, the air temperature in the apparatus reached 38–42°C), and in the afternoon between 13:00 and 15:00 (supposed to be the high-level sun exposure, the air temperature in the apparatus reached 55–60°C). The experimental apparatus was a lidless long box (length×width×height: 40×10×12 cm) (Fig. 2A). One end of the box was for releasing spiders, and the other end was covered by brown paper and provided a shaded sheltering space for spiders to avoid direct sun exposure in the experiments. We made this apparatus by gluing four pieces of white plastic board (two pieces 40×12 cm as the long walls, two pieces 10×12 cm as the short walls) to a white plywood board (40×10 cm as the bottom), then we glued seven narrow plastic boards at equidistant intervals (5.7 cm) along the bottom as obstacles for spiders in the experiments. The seven obstacles increased in height from 1 cm at the first obstacle to 1.5 cm at the sixth obstacle in increments of 0.1 cm, and to 2.5 cm at the seventh obstacle. In the releasing area we put a plastic cup (diameter: 1.5 cm×high: 1 cm) in the middle as the hold space to acclimate the spiders during the experiment. A thin string was connected to the cover of the cup allowing us to open the cup to release the spider.

Each spider was placed into the cup for 1 min to acclimate then the cover was lifted up by pulling the string. We recorded the time each spider took to reach the shelter using a stopwatch. Individuals that failed to reach the shelter within 60 s were recorded as 'failed'. We ran 50 trials (25 for ESC mothers and 25 for ESR mothers) under medium-level sun exposure, and 97 trails (49 for ESC mothers and 48 for ESR mothers) under high-level sun exposure, each spider was tested only once and individuals were randomly allocated to either ESC or ESR group; trials were conducted in an alternating sequence (ESC, ESR, ESC, ESR…) to eliminate any potential order or time-of-day bias.

### Flood

We used a thermostat water bath (DRHH-2, water tank size: length×width×depth: 30×15×20 cm) (Fig. 2B) to test our spiders' escape behavior in 50°C water from one end of the tank to the plastic 'island' (2.5 cm diameter×1 cm height above the water) placed in the other end. This experiment was designed to simulate a flood event, which is extremely common in the study area, particularly during the rainy season. Each test started once the spider was released into the water, the time that each spider used to reach the island was recorded. Individuals that failed to reach the island in 30 s were recorded as 'failed'. We tested 36 ESC mothers and 28 ESR mothers, and each spider was tested once and spiders from each group was tested alternately one by one.

### Predator

We used the same apparatus as in the sun exposure experiment to test our spiders' escape behavior facing predation risks in the laboratory

(humidity: 70–80%, temperature: 25±1°C). To do the test, we fixed two apparatus together side by side along the long wall, then one ESC mother was released into the cup of one apparatus and one ESR mother was released into the cup of the other apparatus at the same time. After 1 min acclimation, the two cups were lifted up simultaneously, at the same time we started to regularly tap the middle of the two walls of the two releasing points with a tiny wooden hammer to mimic the approach of a potential predator (such as a birds or lizards). The time that each spider took to reach the shelter was recorded. Individuals that failed to reach the shelter in 60 s were recorded as 'failed'. We tested 25 ESC mothers and 25 ESR mothers, and each spider was tested only once.

## Data analysis

### Egg sac expansion flexibility and species specificity of mother brooding behavior

To confirm that brooding process is vital for egg hatching and hatchlings' emergence from egg sacs, and to test whether egg sac expansion is a stereotyped behavioral process or demonstrates behavioral flexibility, and whether the spider egg sac expansion behavior is species-specific, we (1) summarized the ratio of egg sacs opened in the four groups, as well as the number and ratio of egg sacs opened on time (as if egg sacs were not opened at the proper time, the hatchlings could not climb onto the back of brooders). (2) We then compared the two ratios above among groups by Fisher's exact tests followed by pairwise comparison with *P*-values adjusted by "holm" method. (3) We calculated the ratio of hatched eggs in each egg sac and summarized the ratios of each group of egg sacs and (4) then compared the hatching possibilities of eggs in sacs brooded by different brooders with a BLMM. As the number of hatched and unhatched eggs could not be counted for un-brooded egg sacs (as the hatched and unhatched embryos could not be separated in this situation), they were not included in 3 and 4.

### Mothers escaping from risks

To test whether ESC females differed from ESR females in risk avoidance, we (1) compared the ratio of successfully escaped spiders in ESC and ESR spiders under different stresses (sun exposure, 'flood' and 'predator'); (2) compared the escaping speeds (distance of the testing arena/time used to reach the safety site) of the two groups of spiders under the three tested conditions. The ratios of successfully escaped spiders in the two groups of spiders were compared by Fisher's exact tests, and the escaping speeds were compared by Bayesian linear models (BLMs). As all the spiders successfully escaped under medium-level sun exposure and 'predator' threatening, and two individuals (ca. 4%) failed in the high-level sun exposure in both groups of spiders, we only compared the successful rate of spiders tested under 'flood'.

All data visualizations and analysis were done in R 4.4.3 through RStudio 2024.12.1 Build 563 (Posit Team, 2025; R Core Team, 2025). The package "ggplot2" (Wickham, 2016) was used for data visualizations. Fisher's exact tests were done with the pairwise.fisher.test function in the package "fmsb" (Nakazawa, 2024). BLMM and BLMs were fitted with the package "brms" (Bürkner, 2017), which uses Stan to perform MCMC sampling (Stan Development Team, 2025). The default priors in "brms" were used and posteriors were estimated across four independent chains, with 10,000 iterations of sampling and a warmup of 1000 per chain. The remaining iterations were thinned to retain every tenth iteration, resulting in a total of 36,000 posterior draws. The settings used for the MCMC algorithm were an adapt delta value (target acceptance rate) of 0.92 and a maximum tree depth of 12. Posterior convergence of each model was diagnosed by visual inspection of trace and density plots, and observation of Rhat values (the potential scale reduction factor) below 1.01 for all estimated parameters (Gelman, 2014). Before interpreting models, we also ensured that all the bulk and tail effect sample sizes were >1000 and the model fits well [checked by pp_check function in "brms" and dh_check_brms function in the package "DHARMa.helpers" (Rodríguez-Sánchez, 2024)]. The evidence for differences and their estimated sizes were calculated from the posterior distributions of parameter estimates in the package "emmeans" (Lenth, 2025). Then we reported median effect sizes (β) and the 95% confidence intervals for highest posterior distribution (CI-HPD). Effects were considered credibly significant when the 95% CI-HPD did not include zero.

Biology Open

## Acknowledgements
We sincerely appreciate Benjamin Blanchard for English editing; Richard T. Corlett for revisions and comments; Guogang Li for assistance in data collection; Yun Fu, Li Wang, Kailun Hu, Xiaodong Yang and Mareike Roeder for experimental equipment assistance; Central Laboratory in XTBG for spider rearing and experimental environments offering.

## Competing interests
The authors declare no competing or financial interests.

## Author contributions
Conceptualization: J.-X.L., Z.C.; Data curation: B.-L.C., J.-X.L., Z.C.; Formal analysis: B.-L.C., J.-X.L.; Funding acquisition: Z.C.; Investigation: Z.C.; Methodology: B.-L.C., J.-X.L., Z.C.; Project administration: Z.C.; Resources: B.-L.C., Z.C.; Software: J.-X.L.; Supervision: Z.C.; Validation: J.-X.L.; Visualization: B.-L.C., J.-X.L., Z.C.; Writing – original draft: Z.C.; Writing – review & editing: B.-L.C., J.-X.L., Z.C.

## Funding
This work was supported by the Hundred Talents Program of Chinese Academy of Sciences (292022000040) and the Ten Thousand Talent Plans for Young Top-notch Talents of Yunnan (20200000099). Open Access funding provided by Chinese Academy of Sciences. Deposited in PMC for immediate release.

## Data and resource availability
All relevant data and details of resources can be found within the article and its supplementary information.

## Peer review history
The peer review history is available online at https://journals.biologists.com/bio/lookup/doi/10.1242/bio.062232.reviewer-comments.pdf

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
