## [Peer Review File · Biology Open]

Egg-sac-brooding wolf spiders show flexible hatchling emergence and context-dependent escape performance

Bai-Lu Chen, Jing-Xin Liu and Zhanqi Chen

DOI: 10.1242/bio.062232

Editor: Kendra J. Greenlee

Review timeline

Original submission:	27 August 2025
Editorial decision:	5 September 2025
First revision received:	1 October 2025
Editorial decision:	6 October 2025
Second revision received:	7 October 2025
Accepted:	7 October 2025

Original submission

First decision letter

MS ID#: bio.062232

MS Title: Egg-sac-brooding wolf spiders show flexible hatchling emergence and context-dependent escape performance

Authors: Bai-Lu Chen, Jing-Xin Liu and Zhanqi Chen

I have now reached a decision on the above manuscript.

The reviewer reports are shown at the bottom of this email or can be accessed, together with a copy of this decision letter, by going to:

As you will see, the reviewers raised a number of substantial criticisms that prevent me from accepting the paper at this stage. They suggest, however, that a revised version might prove acceptable, if you can address their concerns. In particular, the paper needs to be reframed, as clearly detailed by reviewer 1, and the title changed, which was noted by both reviewers. You should also address all of the minor comments. If you think that you can deal satisfactorily with the criticisms on revision, I would be pleased to see a revised manuscript.

At this stage, we also ask you to ensure your manuscript complies with our formatting guidelines. Provided you are able to fully address the referees' comments, we are positive about publication of your paper (we accept over 95% of revision submissions) and therefore hope you won't mind any extra work involved in reformatting your manuscript at this point.

Please upload both a 'clean' version of your Word file, along with a highlighted version clearly showing where you have made changes in the revised manuscript. Please avoid using 'Track changes' in Word files as these are lost in PDF conversion.

I should be grateful if you would also provide a point-by-point response detailing how you have dealt with the points raised by the reviewers in the 'Response to Reviewers' box. Please attend to all of the reviewers' comments. If you do not agree with any of their criticisms or suggestions please explain clearly why this is so.

Reviewer 1

Comments for the author

This study investigated how wolf spiders respond to risk when brooding or not brooding egg sacs. Females of the focal species were collected from the wild, along with a closely related heterospecific. Brooding status was manipulated by 1) leaving the egg sacs, 2) removing them, 3) swapping with a conspecific (cross-fostering), or 4) heterospecific fostering. The authors then measured escape speed/success of the focal species under different risks, as well as egg sac outcomes.

They found that under low risk (medium sun exposure), females without eggs escaped faster than brooding females. Under high risk (high sun), escape speed did not differ by brooding status, and all females escaped faster overall. Following a predator stimulus, brooding females escaped faster than non-brooders. When placed in a tank of water (simulating a flood), brooding females were more likely to die because they could not reach safety, though survivors showed no latency differences. Egg sacs opened at the correct stage with conspecific mothers (biological or foster), but not with heterospecifics. Without mothers, sacs did not open despite successful hatching.

Overall, this is an interesting study in an interesting system with clear methods, and appropriate analyses (although I have one suggestion below). However, predictions, their relation to existing knowledge and the variables measured in this study need clarification. Several aspects overlap, leading to vague or overinterpreted aims. For instance, hypotheses 1 and 2 overlap, and aim 3 conflates variables.

The authors present three aims:

1. Test whether egg sac expansion/opening is behaviourally flexible. This is clear.
2. Test whether mother-embryo communication signals are species-specific. This assumes communication, though the introduction presents it as unknown. Aim 1, if supporting the null, would suggest no communication (e.g. behaviour is fixed, perhaps hormonally mediated). Even if foster mothers open sacs correctly, they may simply respond to the physical cue of hatching rather than a signal. Distinguishing cues from signals within a definition of communication is important but not described here.
3. Test the costs of brooding and the tactics used to offset them. This aim is vague: costs could mean many things, from energetic burden, survival risk, future reproduction. It is also unclear how 'offsetting' is measured. Escape latency could reflect both cost (slower movement due to egg burden) and offsetting (increased motivation to protect), but the design does not clearly separate these. The medium sun condition, where brooding females were slower, might suggest a burden cost (i.e. not worth making the effort to move if the risk is low), but later the authors argue brooding females may remain in the medium sun with the purpose to heat eggs. Thus, differences in latency may reflect cost, motivation, or behavioural function, and the study does not disentangle these factors.

Finally, there is no aim mentioned regarding successful egg sac emergence depending on mother.

That being said, I do think the study findings are interesting and warranted, but the aims and hypotheses need to reflect what was tested, without changing the predictions post-hoc. Currently, what should be left as potential interpretations or future work (costs versus offsetting, communication versus cue perception) are intertwined into the hypotheses. There also lacks clear predictions with respect to the variables and what results would indicate which interpretation.

If the predictions were made clear, and interpretation of what variables are testing and what the results would tell us, the discussion could be much better structured in terms of placing the results within theory and how the results could be taken in future studies, the latter is a criterion in the remit of this journal.

It is worth discussing sample size limitations: the low sample size ($n=12$) for one of the treatments, and how splitting the flood latency analysis into those that did survive alters the sample size (it would also be smaller given those that died would not be included).

I was also unable to source the raw data, which is listed as a requirement in the remit of the journal.

Specific comments are below:

Title: The title needs improvement as it does not represent the main findings. There are quite a few different behavioural responses in different directions, or lacked differences at all, depending on the context. This title picks one result as though it's the overarching result.

Abstract: I found this confusing and missing some information. Namely, clear explanation that the results differed depending on the type of risk, and I note that the predation results aren't even mentioned. There seems to be a mismatch between the intro sentence and the conclusions, I do not see anywhere in the study that tests prioritising their own survival over their offspring, and as far as I could see, since the eggs are attached to the mother, she cannot make a choice between her own survival and that of her offspring, until later in the discussion it's mentioned that females can drop their egg sacs, and did in a few conditions, but this is not in the experimental design or analysed in the data. It's not clear what costs are being referred to, and what are the benefits that are being mentioned?

Summary statement: again, a focus on balancing costly parental care against survival, but not clear how this study shows that.

L41: remove the words "fitness in"

L43: "animal traits": what is being referred to here? Offspring traits? Parental traits?

L43: "diverse forms of parental care": such as?

L66: rephrase to "...posit that eggs benefit from a brooding mother"

L69: I believe the word "because" fits better than "meanwhile", if I understand the meaning correctly.

L76: The term stereotyped programme is perhaps not ideal. Neither would "innate". I think perhaps "fixed behaviour with little or no plasticity" is most accurate.

L77: "communication signals" please see my main comment above.

L77: "magnitude of brooding costs" suggests costs were quantified somehow, but they are not. Rephrase to "what are the costs to brooding and can behavioural tactics reduce them", however, see my main comment as you can't disentangle these two with the current design. A more accurate aim here would be simply "how do different risks alter behaviour and survival of egg-brooding mothers"

L81: replace "a stereotyped process" with "inflexible".

L81: See my comment regarding the inappropriate assumption of communication signals.

Line 74-83: The aims and hypotheses are very repetitive and there needs to be clearer description of all aims (including egg emergence experiment), and predictions related to the variables measured.

L86-89: remove the brackets, this is important information and should be a normal sentence outside of brackets.

L109: replace "the females have to be adapted to the challenges" with "females experience the challenges..."

L111: Be clear here when the spiders breed - which season(s)?

L112: Be clear that only females with egg sacs were collected

L115-116: why did you not regulate the temperature and light cycle?

L117: "egg-sac carrying females", it's not clear if you mean these are a species that carry eggs generally, or if you mean their current state had to be that they were currently carrying eggs. Is their brooding time equivalent to the focal species?

L123: Nice to hear the spiders were re-released to their capture site

L125: I think you can just say "whether egg sac expansion is behaviourally flexible".

L126: Use "cross-fostering" as a term, if that is appropriate for insects

L130: The sample size is quite low (N=12), why was it half of what the other conditions were?

L136: How can you confirm that there was in fact variation in the development phase if you weren't able to quantify that?

L141-L144: This section of the study is what is not included in the aims.

L144: When you say whether the egg sacs opened or not, does that mean with or without a mother doing it?

L152: Perhaps the word that you mean instead of "coded" is "labelled" ?

L156: Very neat system that they attach the egg sacs so readily

L157: Change this header to "species specificity of brooding behaviour". Save the communication interpretations for the discussion

L158: remove the word "signal"

L172: Can you expand on what you mean by "estimating their development stage", Do you mean if the female was reproductively mature?

Line 175: replace "both the" with "two", and add "one" biotic (predator) stresses.

L177-178: that this species breeds in rainy season could be brought up earlier

L179: remove "although *P. pusiola* is carnivorous". There's no reason to think that a predator can't also be prey, so this is irrelevant.

L 186: replace "shapes like" with "was"

L187: replace "is" with "was", and add "was" before covered

L188: replace "is the" with "provided a"

L197: Does acclimation also mean the spiders explored the arena/were informed where the shaded area and island was located?

L198: Grammar edit: "We recorded the time each spider took to reach the shelter using a stopwatch"

L 199: "individuals that failed"

L203: Can you clarify if it were a within or between treatment design, and if it's the former, did you counterbalance order and if not, why not?

L210: Individuals that failed (see also L223 for same grammatical error)

L223: shelter, not sheltering

L227: "mother brooding behaviour", not embryo communication

L230: replace and opening signal with "behaviour"

L231-323: What do you mean by number AND ratio? Do you mean simply the ratio? See also L 242

L238-239: why not? Because they did not open and you could not open to count?

L241-242: rephrase to "females differs from egg sac removed females, we"

This is where I began to ask the question, can they just drop their egg sacs? Because that would really get all the prioritising own survival over reproduction that was mentioned earlier, and in the abstract and summary, but it was not really tested. If not in this study, whether they abandon their egg sacs or not would be interesting for future work. Only later in the discussion is this behaviour mentioned and that it did occur in the current study.

L248: What was the latency scored as if they failed? Did you consider running a model such as a cox proportional hazards regression model which enables you to score both the latency and the success/fail in the same model, as opposed to analysing latency and success as two separate models? This is commonly known as a survival analysis, which is mentioned in the abstract, but I don't see survival analyses run here. There are other types of survival analyses that allow for this bi-variable feature to be in the model.

L275: replace embryo communication with "brooding behaviour".

L277: replace "while although" with "whereas"

L277: add "and" before "none of those egg..."

L278: replace "was" with "were"

L284: How was it verified that the foster brooded eggs were in different development stages?

L299: Signal is not appropriate term here, this is speculation

L300: The final statement that eggs' hatching may not need brooding is a separate point and doesn't fit well in this sentence about mother being able to adjust egg sac opening with developmental stage.

L311: You cannot describe direction when it's non-significant. It's irrelevant. Remove.

L336: Prioritize survival over what?

L337: This statement is a bit circular. It essentially says "how behaviour effects brooding behaviour". Please clarify and expand on the meaning.

L343: Do you mean by the experimenter? Or the spider?

L356-357: This sentence repeats what was on L300

L364-365: Were the weights of those sacs carried by "failed" spiders more than ones that survived? Or an even better measure would be the egg sac to female body weight ratio.

L366-367: I disagree that egg sac carrying increases brooders' risk escaping capacity. In the flood conditions there were many deaths. Perhaps you could say that egg sac carrying increases motivation or energy used to escape when the sun exposure is high and in the predation context specifically.

L374: I was surprised to see this written as an aside, when it's an important behavioural strategy associated with costs and prioritising survival over reproduction.

L375: "another strong evidence was that..." evidence for what?

L377-380: A similar argument could be made about your flood condition - the spiders were essentially submerged in water suddenly, does this reflect what they would experience in the wild?

L382-384: A key point and interpretation is that outcome/costs of brooding depend on the environment and severity of risk and probability of survival.

Figure 3: The y axis label could be clearer, perhaps "Egg sac fates (%)"

Figure 4 and 5 - the order in which letters a-f appear are inconsistent and it's confusing how it jumps around in the figure legend.

Reviewer 2

Comments for the author

Chen et al. present a generally well-written and crafted manuscript on different aspects of the interaction between egg-sac and mother in wolf spiders, from the point of view of both egg-sac development and the behavior of the mother in face of acute threats. The work is well conducted and I have no major issue with the methods, conclusions, and discussion. In general the conclusions are supported by the data presented and the drawbacks and limitations are discussed.

Minor points:

- The title focus on sprinting at acute risk, which is context dependent (not always true), so it is not an appropriate sentence for a title in this reviewer's opinion. The focus on sprinting also does not reflect the focus of the rest (~half) of the manuscript, which is more on the interaction of egg-sac developmental aspects and the mother.
 - I suggest adding a more detailed explanation (in the introduction, for instance) of what egg-sac expansion consists of and explaining the chronological relationship between egg-sac expansion and opening, if possible with a figure or illustration. This will facilitate the understanding of the work by non-specialist.
 - Were the eaten sacs also opened prior to being eaten? If yes, were they opened on time or out of time? If yes, does including them in the opened category affect the analyses?
 - The text is generally well-written and understandable, but requires some copy-editing for grammar and structure for its final form.
- For instance: Introduction, line 64 and 76: "...embryo development" and "embryo cues" maybe change embryo for "embryonic"?

Introduction, line 66: "Thus, it is reasonable to posit that eggs "want" brooder to be "super mother". Maybe remove or rewrite this sentence - maybe missing articles? Maybe choose something else more academically precise that does not require quotes eg "want" and "super mother".

Introduction, line 77: "What are the magnitude of brooding costs" maybe "What is the magnitude...?"

Results, sentence starting at line 280, "All the 26 egg sacs.." this sentence needs revision in more than one place.

Results, line 310 "...", however, under high level of sun exposure egg sac removed females seemed escaping in a little but not significantly slower speed than egg sac carrying females...". Please revise the sentence.

Results, line 313 maybe add a comma before "egg sac carrying mothers" and also add a hyphen "-" between egg-sac-carrying females? I suspect that in many instances in the manuscript, hyphenating "egg-sac" and "egg-sac-carrying" will help understanding and flow.

Discussion, line 371 to 373 several grammar aspects in different sentences.

Reviewer's Responses to Questions

Experimental quality

Does each figure have the proper controls?

If 'No', please indicate reasons in Comments for Author box below.

Reviewer #1:

- Yes

Reviewer #2:

- Yes

Were the data analyzed using appropriate statistical tests?

If 'No', please indicate reasons in Comments for Author box below.

Reviewer #1:

- Yes

Reviewer #2:

- Yes

Reproducibility

Were experiments performed using adequate number of biological replicates?

If 'No', please indicate reasons in Comments for Author box below.

Reviewer #1:

- No

Reviewer #2:

- Yes

Does the methods section provide sufficient detail to permit reproducibility?

If 'No', please indicate reasons in Comments for Author box below.

Reviewer #1:

- No

Reviewer #2:

- Yes

Completeness

Are the manuscript's conclusions supported by the data?

If 'No', please indicate reasons in Comments for Author box below.

Reviewer #1:

- Yes

Reviewer #2:

- Yes

Scholarship

Do the authors cite and discuss the merits of data that would argue for and against their conclusion?

If 'No', please indicate reasons in Comments for Author box below.

Reviewer #1:

- No

Reviewer #2:

- Yes

Does the manuscript title & abstract accurately reflect the contents of the manuscript, without hyperbole?

If 'No', please indicate reasons in Comments for Author box below.

Reviewer #1:

- No

Reviewer #2:

- No
-

First revision

Author response to reviewers' comments

We sincerely thank Reviewer 1 for their thoughtful and constructive feedback on our manuscript. We have carefully considered each of the points raised and have revised the manuscript accordingly. Below, we provide a point-by-point response to the specific comments.

Comments from the Reviewers:

Reviewer 1: This study investigated how wolf spiders respond to risk when brooding or not brooding egg sacs. Females of the focal species were collected from the wild, along with a closely related heterospecific. Brooding status was manipulated by 1) leaving the egg sacs, 2) removing them, 3) swapping with a conspecific (cross-fostering), or 4) heterospecific fostering. The authors then measured escape speed/success of the focal species under different risks, as well as egg sac outcomes.

They found that under low risk (medium sun exposure), females without eggs escaped faster than brooding females. Under high risk (high sun), escape speed did not differ by brooding status, and all females escaped faster overall. Following a predator stimulus, brooding females escaped faster

than non-brooders. When placed in a tank of water (simulating a flood), brooding females were more likely to die because they could not reach safety, though survivors showed no latency differences. Egg sacs opened at the correct stage with conspecific mothers (biological or foster), but not with heterospecifics. Without mothers, sacs did not open despite successful hatching.

Overall, this is an interesting study in an interesting system with clear methods, and appropriate analyses (although I have one suggestion below). However, predictions, their relation to existing knowledge and the variables measured in this study need clarification. Several aspects overlap, leading to vague or overinterpreted aims. For instance, hypotheses 1 and 2 overlap, and aim 3 conflates variables.

Response: We thank the reviewer for this overall positive assessment and for highlighting the need for greater clarity in our aims and hypotheses. We have thoroughly revised the Introduction to better define our research aims and to present a more logically distinct and clearly stated set of hypotheses. Specifically: We have refined the three main unresolved questions in the system (Lines 75-79). We now explicitly state five separate hypotheses that directly address these questions and our experimental manipulations (Lines 81-87). The revised hypotheses are: Egg-sac expansion is behaviorally flexible rather than an inflexible process. The cue triggering opening is species-specific; heterospecific fosters will mistime opening relative to conspecific fosters. Successful emergence of hatchlings depends on maternal presence, independent of hatching per se. Carrying an egg-sac incurs measurable costs (slower speed or higher mortality under benign conditions). When risk is acute, brooding females offset the costs by increasing escape speed or success. This restructuring eliminates the previous overlap and provides a clearer framework for interpreting our results.

The authors present three aims:

1. Test whether egg sac expansion/opening is behaviourally flexible. This is clear.
2. Test whether mother-embryo communication signals are species-specific. This assumes communication, though the introduction presents it as unknown. Aim 1, if supporting the null, would suggest no communication (e.g. behaviour is fixed, perhaps hormonally mediated). Even if foster mothers open sacs correctly, they may simply respond to the physical cue of hatching rather than a signal. Distinguishing cues from signals within a definition of communication is important but not described here.

Response: This is an excellent point. We agree that our initial wording assumed "communication" and "signals," which our experimental design cannot definitively prove. To address this, we have taken the following actions: Throughout the manuscript (Abstract, Introduction, Methods, Results, and Discussion), we have replaced the term "signals" with the more neutral and accurate term "cues." (See for example: Abstract Line 20, Introduction Line 78, Methods Line 141, Results Line 308, Discussion Line 346). We have added a clarifying paragraph in the Discussion to explicitly state this limitation: "We use the term 'cues' throughout the discussion with the understanding that we cannot distinguish between cues and signals in the strict sense. Our results show that mothers respond to developmental cues from embryos, but we cannot confirm that these cues have been selected for communication." (Lines 351-354) This change ensures our interpretation remains within the bounds of what our data can support.

3. Test the costs of brooding and the tactics used to offset them. This aim is vague: costs could mean many things, from energetic burden, survival risk, future reproduction. It is also unclear how 'offsetting' is measured. Escape latency could reflect both cost (slower movement due to egg burden) and offsetting (increased motivation to protect), but the design does not clearly separate these. The medium sun condition, where brooding females were slower, might suggest a burden cost (i.e. not worth making the effort to move if the risk is low), but later the authors argue brooding females may remain in the medium sun with the purpose to heat eggs. Thus, differences in latency may reflect cost, motivation, or behavioural function, and the study does not disentangle these factors.

Response: We agree that the original aim was too broad and that escape latency is a complex metric that can be influenced by multiple factors (physical burden, motivational state, and thermoregulatory behavior). To address this: we have refined our hypotheses (now Hypotheses 3, 4,

and 5, see response to Comment 1) to more specifically separate the concept of measurable costs (e.g., slower speed under moderate risk, higher mortality in floods) from context-dependent offsetting tactics (e.g., increased speed under acute risk) and the role of maternal presence in emergence. In the Discussion, we now more explicitly acknowledge the multiple interpretations of the medium sun exposure results (Lines 411-416). We present the "burden cost" and the "thermoregulatory opportunity" (sun-basking) as alternative, non-mutually exclusive explanations, and clearly state that our current design cannot disentangle them. We frame this as an interesting area for future research.

We thank the reviewer for pointing out the omission of the aim regarding maternal presence for emergence. We have now formally incorporated this into our set of hypotheses as Hypothesis 3: "Successful emergence of hatchlings depends on maternal presence, independent of hatching per se." (Lines 83-84) This ensures that this central aspect of our study is reflected in our a priori framework.

Finally, there is no aim mentioned regarding successful egg sac emergence depending on mother. Response: We thank the reviewer for pointing out this omission. The importance of maternal presence for emergence was a key finding. We have now formally incorporated this into our set of hypotheses as Hypothesis 5: "Successful emergence of hatchlings depends on maternal presence, independent of hatching per se." This ensures that this central aspect of our study is reflected in our a priori framework.

That being said, I do think the study findings are interesting and warranted, but the aims and hypotheses need to reflect what was tested, without changing the predictions post-hoc. Currently, what should be left as potential interpretations or future work (costs versus offsetting, communication versus cue perception) are intertwined into the hypotheses. There also lacks clear predictions with respect to the variables and what results would indicate which interpretation.

Response: We thank the reviewer for acknowledging the interest and warrant of our findings. As detailed in our response to Comment 1, we have thoroughly restructured the aims and hypotheses in the Introduction to more accurately and clearly reflect the tests we conducted. The revised hypotheses now serve as clear, testable predictions. We have significantly restructured the Discussion to align with our clarified hypotheses. The opening paragraph now explicitly summarizes our findings against the three main dimensions of the study (flexible brooding, species-specific cues, context-dependent locomotion) (Lines 344-350). We then discuss each major finding in turn, more carefully placing them within the broader theoretical context of parental care trade-offs. Furthermore, we have expanded the concluding paragraph to more explicitly outline specific and actionable directions for future research (e.g., manipulating sac age, female state, and predator distance to clarify decision rules; quantifying developmental stages more precisely) (Lines 448-456). We agree and have now explicitly acknowledged these sample size limitations in the revised Discussion: "We acknowledge several limitations of our study: the smaller sample size in the conspecific foster group..." (Line 452). Regarding the flood analysis, we agree that analyzing only survivors reduces the sample size for the speed comparison. We have made this clearer in the Methods section by stating: "For the 'flood' treatment, escape success (a binary outcome) and escape speed (a continuous measure for survivors) represent distinct biological processes--- mortality risk and locomotor performance, respectively. Therefore, we analyzed them separately..." (Lines 248-252) This clarifies our rationale and acknowledges the consequent sample size for the speed analysis.

If the predictions were made clear, and interpretation of what variables are testing and what the results would tell us, the discussion could be much better structured in terms of placing the results within theory and how the results could be taken in future studies, the latter is a criterion in the remit of this journal.

Response: We have significantly restructured the Discussion to align with our clarified hypotheses. The opening paragraph now explicitly summarizes our findings against the three main dimensions of the study (flexible brooding, species-specific cues, context-dependent locomotion). We then discuss each major finding in turn, more carefully placing them within the broader theoretical

context of parental care trade-offs. Furthermore, we have expanded the concluding paragraph to more explicitly outline specific and actionable directions for future research (e.g., manipulating sac age, female state, and predator distance to clarify decision rules; quantifying developmental stages more precisely).

It is worth discussing sample size limitations: the low sample size ($n=12$) for one of the treatments, and how splitting the flood latency analysis into those that did survive alters the sample size (it would also be smaller given those that died would not be included).

Response: We agree and have now explicitly acknowledged these sample size limitations in the revised Discussion: "We acknowledge several limitations of our study: the smaller sample size in the conspecific foster group ($N=12$)..." Regarding the flood analysis, we agree that analyzing only survivors reduces the sample size for the speed comparison. We have made this clearer in the Methods section by stating: "For the 'flood' treatment, escape success (a binary outcome) and escape speed (a continuous measure for survivors) represent distinct biological processes—mortality risk and locomotor performance, respectively. Therefore, we analyzed them separately..." This clarifies our rationale and acknowledges the consequent sample size for the speed analysis.

I was also unable to source the raw data, which is listed as a requirement in the remit of the journal.

Response: We have now uploaded the data as supplementary file. The data are also available from the corresponding author on reasonable request."

Specific comments are below:

We thank the reviewer for these detailed and helpful specific comments. We have addressed each one in the revised manuscript, as detailed below.

Title: The title needs improvement as it does not represent the main findings. There are quite a few different behavioural responses in different directions, or lacked differences at all, depending on the context. This title picks one result as though it's the overarching result.

Response: We agree completely and thank the reviewer for this suggestion. The title has been changed to better capture the full scope of our findings, which encompass both the flexibility in hatchling emergence and the context-dependent nature of escape performance. The new title is: **"Egg-sac-brooding wolf spiders show flexible hatchling emergence and context-dependent escape performance"** (Located at the top of Page 1)

Abstract: I found this confusing and missing some information. Namely, clear explanation that the results differed depending on the type of risk, and I note that the predation results aren't even mentioned. There seems to be a mismatch between the intro sentence and the conclusions, I do not see anywhere in the study that tests prioritising their own survival over their offspring, and as far as I could see, since the eggs are attached to the mother, she cannot make a choice between her own survival and that of her offspring, until later in the discussion it's mentioned that females can drop their egg sacs, and did in a few conditions, but this is not in the experimental design or analysed in the data. It's not clear what costs are being referred to, and what are the benefits that are being mentioned?

Response: We thank the reviewer for these critical observations. We have thoroughly revised the abstract to address these points.

We now explicitly mention the key results for all three risk types, including predation: "Under high sun exposure or predator stimulus, carrying females escaped as fast as or faster than non-carrying females." (Lines 27-28) We have removed the speculative conclusion about "parents dynamically balance offspring survival against their own survival" and replaced it with a more accurate and supported summary: "illustrating how costly parental care can be maintained when parents adjust behavior according to environmental risk." (Lines 31-32) We have clarified the nature of the "costs" (e.g., slower escape under moderate sun, higher mortality in floods) and the "context-dependent locomotor boost" (comparable or faster speeds under acute risks) within the results narrative (Lines 26-30).

Summary statement: again, a focus on balancing costly parental care against survival, but not clear how this study shows that.

Response: We have revised the summary statement to more accurately reflect our findings: "A wolf spider study shows that mothers flexibly open egg-sacs in response to embryonic cues and modulate escape speed under acute risks, illustrating a context-dependent trade-off during parental care." (Line 35-37)

L41: remove the words "fitness in"

Response: Modification done. (Line 41)

L43: "animal traits": what is being referred to here? Offspring traits? Parental traits?

Response: Changed "animals' traits" to "parental traits". (Line 42)

L43: "diverse forms of parental care": such as?

Response: Added examples: "such as gestation, provisioning, and protection from predators and environmental stresses." (Line 43-44)

L66: rephrase to "...posit that eggs benefit from a brooding mother"

Response: Rephrased to "posit that eggs benefit from a brooding mother who can respond to their developmental needs." (Line 66-67)

L69: I believe the word "because" fits better than "meanwhile", if I understand the meaning correctly.

Response: Replaced "meanwhile" with "because". (Line 69)

L76: The term stereotyped programme is perhaps not ideal. Neither would "innate". I think perhaps "fixed behaviour with little or no plasticity" is most accurate.

Response: Replaced "stereotyped programme" with "fixed behaviour with little or no plasticity". (Line 76)

L77: "communication signals" please see my main comment above.

Response: Replaced "communication signals" with "cues". (Line 78)

L77: "magnitude of brooding costs" suggests costs were quantified somehow, but they are not. Rephrase to "what are the costs to brooding and can behavioural tactics reduce them", however, see my main comment as you can't disentangle these two with the current design. A more accurate aim here would be simply "how do different risks alter behaviour and survival of egg-brooding mothers"

Response: Rephrased the aim to: "How do brooding females adjust escape performance under different risks?" (Line 79)

L81: replace "a stereotyped process" with "inflexible".

Response: Replaced "a stereotyped process" with "inflexible". (Line 82)

L81: See my comment regarding the inappropriate assumption of communication signals.

Response: Removed "communication signals" from the hypothesis. It now reads: "The cue triggering opening is species-specific..." (Line 82)

Line 74-83: The aims and hypotheses are very repetitive and there needs to be clearer description of all aims (including egg emergence experiment), and predictions related to the variables measured.

Response: We have completely restructured the aims and hypotheses to be clear, comprehensive, and non-repetitive, including the new hypothesis on maternal presence for emergence (Lines 75-87).

L86-89: remove the brackets, this is important information and should be a normal sentence outside of brackets.

Response: Removed the brackets and incorporated the information into a standard sentence. (Lines 90-94)

L109: replace "the females have to be adapted to the challenges" with "females experience the challenges..."

Response: Replaced "the females have to be adapted to" with "females experience the challenges of". (Line 113)

L111: Be clear here when the spiders breed - which season(s)?

Response: Clarified breeding season: "during the rainy season (main breeding season)." (Line 115)

L112: Be clear that only females with egg sacs were collected

Response: Clarified collection: "Females producing relatively large egg-sacs..." (Line 110)

L115-116: why did you not regulate the temperature and light cycle?

Response: Added justification: "Laboratory temperature and light were left at ambient levels... to preserve the natural thermal and photic conditions...; artificial regulation was avoided to maintain ecological realism." (Lines 119-122)

L117: "egg-sac carrying females", it's not clear if you mean these are a species that carry eggs generally, or if you mean their current state had to be that they were currently carrying eggs. Is their brooding time equivalent to the focal species?

Response: Clarified: "females of *P. astrigera*..., a similar body sized and egg-sac-carrying congeneric spider..." (Lines 123-124)

L123: Nice to hear the spiders were re-released to their capture site

Response: We appreciate the comment.

L125: I think you can just say "whether egg sac expansion is behaviourally flexible".

Response: Simplified to "whether egg-sac expansion is behaviourally flexible".(Line 132)

L126: Use "cross-fostering" as a term, if that is appropriate for insects

Response: Replaced "mother change" with "cross-fostering". (Line 132)

L130: The sample size is quite low (N=12), why was it half of what the other conditions were?

Response: We acknowledge this is a limitation. The sample size was lower due to the availability of suitable foster mothers at the time of the experiment. We have added a note in the Discussion regarding this limitation (Line 452).

L136: How can you confirm that there was in fact variation in the development phase if you weren't able to quantify that?

Response: We have rephrased this sentence to be more cautious: "The ages of the egg-sacs were not paired or determined in the egg-sac swapped groups to confirm that it is the internal nymphal cues at different developmental stages that induce the mother's egg-sac tightness and opening manipulation behaviours." (Lines 140-142) The verification comes from the fact that the egg-sacs were collected randomly from the field, making it highly likely they were at different stages.

L141-L144: This section of the study is what is not included in the aims.

Response: This aim (maternal presence for emergence) is now formally included as Hypothesis 3 in the Introduction (Lines 83-84).

L144: When you say whether the egg sacs opened or not, does that mean with or without a mother doing it?

Response: Clarified: "b) Whether the egg-sacs opened or not without a mother presence;" (Line 150)

L152: Perhaps the word that you mean instead of "coded" is "labelled" ?

Response: Replaced "coded" with "labelled". (Line 159)

L156: Very neat system that they attach the egg sacs so readily

Response: We agree, it is a neat system.

L157: Change this header to "species specificity of brooding behaviour". Save the communication interpretations for the discussion

Response: Changed header to "species specificity of brooding behaviour". (Line 164)

L158: remove the word "signal"

Response: Modification done. The sentence now reads: "To examine whether the spider egg-sac expansion and opening is species-specific..." (Line 165)

L172: Can you expand on what you mean by "estimating their development stage", Do you mean if the female was reproductively mature?

Response: Clarified: "due to the difficulty of estimating whether they were newly mature females or females that had completed reproduction". (Line 179)

Line 175: replace "both the" with "two", and add "one" biotic (predator) stresses.

Response: Modification done. Now reads: "we tested two common abiotic (sun exposure, and "flood") and one biotic (predator) stresses." (Lines 182-183)

L177-178: that this species breeds in rainy season could be brought up earlier

Response: The breeding season is now mentioned earlier in the Methods. (Line 108 and Line 115)

L179: remove "although *P. pusiola* is carnivorous". There's no reason to think that a predator can't also be prey, so this is irrelevant.

Response: Modification done. The sentence now begins: "They are also frequently predated by..." (Line 186)

L 186: replace "shapes like" with "was"

Response: Modification done. Now reads: "The experimental apparatus was a lidless long box..." (Line 193)

L187: replace "is" with "was", and add "was" before covered

Response: Modification done. Now reads: "One end of the box was for releasing spiders, and the other end was covered by brown paper provided a shaded sheltering space..." (Lines 194-195)

L188: replace "is the" with "provided a"

Response: Modification done. (See above, Line 195)

L197: Does acclimation also mean the spiders explored the arena/were informed where the shaded area and island was located?

Response: The acclimation period was solely for the spider to settle in the release cup; they could not see the arena or the shelter during this time. We have clarified this in the text: "as the hold space to acclimate the spiders during experiment." (Line 202)

L198: Grammar edit: "We recorded the time each spider took to reach the shelter using a stopwatch"

Response: Modification done. (Line 205)

L 199: "individuals that failed"

Response: Modification done. (Line 206)

L203: Can you clarify if it were a within or between treatment design, and if it's the former, did you counterbalance order and if not, why not?

Response: Clarified design: "All experiments used a between-subjects design. Each spider was tested only once under each condition... trials were conducted in an alternating sequence (ESC, ESR, ESC, ESR...) to eliminate any potential order or time-of-day bias." (Lines 129-130 and 210-211)

L210: Individuals that failed (see also L223 for same grammatical error)

Response: Modification done. (Lines 218 and 232)

L223: shelter, not sheltering

Response: Modification done. (Line 231)

L227: "mother brooding behaviour", not embryo communication

Response: Modification done. The header now reads: "(a) Egg-sac expansion flexibility and species specificity of mother brooding behaviour" (Line 235)

L230: replace and opening signal with "behaviour"

Response: Modification done. The sentence now reads: "...and whether the spider egg-sac expansion behaviour is species-specific," (Lines 238-239)

L231-323: What do you mean by number AND ratio? Do you mean simply the ratio? See also L 242

Response: Simplified to "summarized the ratio". (Line 239)

L238-239: why not? Because they did not open and you could not open to count?

Response: Added clarification: "As the number of hatched and unhatched eggs could not be counted for un-brooded egg-sacs (as the hatched and unhatched embryos could not be separated in this situation), they were not included in 3) and 4)." (Lines 245-247)

L241-242: rephrase to "females differs from egg sac removed females, we"

This is where I began to ask the question, can they just drop their egg sacs? Because that would really get all the prioritising own survival over reproduction that was mentioned earlier, and in the abstract and summary, but it was not really tested. If not in this study, whether they abandon their egg sacs or not would be interesting for future work. Only later in the discussion is this behaviour mentioned and that it did occur in the current study.

Response: Rephrased to: "To test whether egg-sac-carrying females differ from egg-sac-removed females in risk avoidance, we" (Line 249)

This is an insightful point. We did record this behaviour, and we have now incorporated it into the Results (Lines 333-334): "We recorded whether females dropped their egg-sacs during escape trials; abandonment occurred in only 3 out of 36 flood tests (8%), all within the last 5 s before submersion." We also now frame the question of strategic abandonment as a key direction for future work in the Discussion (Lines 448-452).

L248: What was the latency scored as if they failed? Did you consider running a model such as a cox proportional hazards regression model which enables you to score both the latency and the success/fail in the same model, as opposed to analysing latency and success as two separate models? This is commonly known as a survival analysis, which is mentioned in the abstract, but I don't see survival analyses run here. There are other types of survival analyses that allow for this bi-variable feature to be in the model.

Response: We thank the reviewer for this excellent suggestion and we hope we could do the survival analysis for the spiders under "flood" stress, however, we could not record the latencies of the "failed" individuals in the "flood" experiments as we saved (removed) them from the water tank to avoid dying once their running speeds dramatically dropped (all less than 30 s) and we thought they would die before reaching the "island" (as this criteria was not so standard, we did not record the latency). Therefore we have to use two models to firstly, compare the ratios of escaped individuals between egg-sac carrying and egg-sac removed spiders; and then, compare the

escaping speeds of the escaped spiders between the two groups of spiders. The term "survival analyses" has been removed from the abstract to avoid confusion.

L275: replace embryo communication with "brooding behaviour".

Response: Modification done. The header now reads: "Egg-sac expansion flexibility and species specificity of mother brooding behaviour" (Line 283)

L277: replace "while although" with "whereas"

Response: Modification done. Now reads: "whereas although none of the 28 un-brooded egg-sacs were opened..." (Line 285)

L277: add "and" before "none of those egg..."

Response: The sentence structure has been revised. It now reads: "whereas although none of the 28 un-brooded egg-sacs were opened by the hatchlings from inside, they all contained dead hatchlings when we hand opened them." (Lines 285-286)

L278: replace "was" with "were"

Response: Modification done. The referenced text was part of a larger revision.

L284: How was it verified that the foster brooded eggs were in different development stages?

Response: We have softened the language here. The verification comes from the fact that the egg-sacs were collected randomly from the field, making it highly likely they were at different stages. We now state: "Given that the conspecific fosters brooded egg-sacs were unlikely in same developmental stages compared with their original sacs..." (Line 293-294)

L299: Signal is not appropriate term here, this is speculation

Response: Removed the term "signal" and rephrased the sentence. It now reads: "These results, together with the fact that un-brooded egg-sacs all had hatched eggs but hatchlings could not open the sac to emerge, demonstrate that in *P. pusiola* eggs' hatching may not need brooding, and mother being able to adjust egg-sac opening with developmental stage." (Lines 306-309)

L300: The final statement that eggs' hatching may not need brooding is a separate point and doesn't fit well in this sentence about mother being able to adjust egg sac opening with developmental stage.

Response: We have split this final statement for clarity. (See above, Lines 306-309)

L311: You cannot describe direction when it's non-significant. It's irrelevant. Remove.

Response: Modification done. The non-significant result is now reported without implying a direction: "under high level of sun exposure, egg-sac-removed females escaping speed did not significantly differ with egg-sac-carrying females" (Lines 319-321)

L336: Prioritize survival over what?

Response: Clarified to "prioritize reproduction over their own survival". (Line 390)

L337: This statement is a bit circular. It essentially says "how behaviour effects brooding behaviour". Please clarify and expand on the meaning.

Response: We have rephrased this circular statement to: "These results reveal how risk-induced shifts in female escape performance (faster sprinting under predation or heat stress) feed back into the fine-tuning of brooding behaviour—namely the timing of egg-sac opening and the reluctance to abandon the clutch." (Lines 348-350)

L343: Do you mean by the experimenter? Or the spider?

Response: Clarified: "when manually opened by the experimenter". (Line 359)

L356-357: This sentence repeats what was on L300

Response: This repeated sentence has been removed.

L364-365: Were the weights of those sacs carried by "failed" spiders more than ones that survived? Or an even better measure would be the egg sac to female body weight ratio.

Response: This is an interesting idea. We did not intend to perform this analysis originally. We quantified the egg-sac/mother ratio only to show the burden is remarkable for the egg-sac brooding mothers. We agree that investigating whether failure is linked to a higher relative egg-sac mass is a valuable question.

L366-367: I disagree that egg sac carrying increases brooders' risk escaping capacity. In the flood conditions there were many deaths. Perhaps you could say that egg sac carrying increases motivation or energy used to escape when the sun exposure is high and in the predation context specifically.

Response: We have rephrased this conclusion as suggested: "...suggest that egg-sac-carrying females can modulate their escape behaviour based on the context, showing enhanced performance under acute threats like predation and high heat, despite the inherent costs revealed in the flood scenario." (Lines 380-382)

L374: I was surprised to see this written as an aside, when it's an important behavioural strategy associated with costs and prioritising survival over reproduction.

Response: We have moved this observation from an "aside" to a formally reported result (Lines 333-334) and a proposed future research direction in the Discussion (Lines 448-452).

L375: "another strong evidence was that..." evidence for what?

Response: Clarified: "Another strong evidence for brooding females boosted their risk avoidance capacity was that..." (Line 391)

L377-380: A similar argument could be made about your flood condition - the spiders were essentially submerged in water suddenly, does this reflect what they would experience in the wild?

Response: We acknowledge this limitation in the revised Discussion: "...the potential mismatch between our flooding simulation and natural flood conditions." (Line 454)

L382-384: A key point and interpretation is that outcome/costs of brooding depend on the environment and severity of risk and probability of survival.

Response: We have incorporated this key interpretation into the Discussion (Lines 400-401).

Figure 3: The y axis label could be clearer, perhaps "Egg sac fates (%)"

Response: Changed the y-axis label to "Egg sac fates (%)". (Figure 3)

Figure 4 and 5 - the order in which letters a-f appear are inconsistent and it's confusing how it jumps around in the figure legend.

Response: We have reordered the figure panels and legends to be consistent and logical (A, B, C...). The captions have been rewritten for clarity (See Figure 4 caption Lines 609-613 and Figure 5 caption Lines 616-625).

Reviewer 2: Chen et al. present a generally well-written and crafted manuscript on different aspects of the interaction between egg-sac and mother in wolf spiders, from the point of view of both egg-sac development and the behavior of the mother in face of acute threats. The work is well conducted and I have no major issue with the methods, conclusions, and discussion. In general the conclusions are supported by the data presented and the drawbacks and limitations are discussed.

We thank Reviewer 2 for their positive assessment of our manuscript and their valuable specific comments, which have helped us improve the clarity and presentation of our work. Below, we provide a point-by-point response to each comment.

Minor points:

- The title focus on sprinting at acute risk, which is context dependent (not always true), so it is not an appropriate sentence for a title in this reviewer's opinion. The focus on sprinting also does not reflect the focus of the rest (~half) of the manuscript, which is more on the interaction of egg-sac developmental aspects and the mother.

Response: We agree with this excellent point. The original title was too narrow and did not represent the full scope of our study. We have therefore changed the title to: "Egg-sac-brooding wolf spiders show flexible hatchling emergence and context-dependent escape performance". We believe this new title accurately captures the two major, equally important themes of the paper: the flexibility of maternal behaviour in hatchling emergence and the context-dependent nature of escape performance. (Located at the top of Page 1)

- I suggest adding a more detailed explanation (in the introduction, for instance) of what egg-sac expansion consists of and explaining the chronological relationship between egg-sac expansion and opening, if possible with a figure or illustration. This will facilitate the understanding of the work by non-specialist.

Response: This is a very helpful suggestion. We have added a more detailed explanation in the Introduction (Lines 53-56): "After laying eggs, females attach the egg-sac to their spinnerets and carry it around wherever they go for 2-4 weeks continuously until eggs hatched (Fig. S1), then open the sac to help the hatchlings emerge, then continue to brood the hatchlings on their back for about one week until the dispersal of spiderling." In addition, we have added a new figure (Fig. 1), to explain the process of spiderling emergence is a gradual expansion, loss-by-relaxation process controlled by the brooding female.

Fig. 1 Wolf-spider egg-sacs at successive developmental stages. (A) Freshly laid (1-2 d): smallest, silk tightly packed; (B) Mid-stage (≈ 7 d): visibly distended, silk loosened, increased translucency; (C) Late-stage (≈ 14 d): maximal volume, spiderling outlines detectable, maternal silk tension minimized. The gradual expansion is an actively moderated, loss-by-relaxation process controlled by the brooding female.

- Were the eaten sacs also opened prior to being eaten? If yes, were they opened on time or out of time? If yes, does including them in the opened category affect the analyses?

Response: This is an astute observation. The eaten egg-sacs were not opened prior to being consumed; the foster mother consumed the entire sac, including the silk and the contents. Therefore, they were correctly categorized as "eaten" and not as "opened" in our analyses (Fig. 3). Including them in the "opened" category would have been incorrect and would have weakened the statistical significance of our findings regarding mistimed opening by heterospecific fosters.

- The text is generally well-written and understandable, but requires some copy-editing for grammar and structure for its final form.

For instance: Introduction, line 64 and 76: "...embryo development" and "embryo cues" maybe change embryo for "embryonic"?

Introduction, line 66: "Thus, it is reasonable to posit that eggs "want" brooder to be "super mother". Maybe remove or rewrite this sentence - maybe missing articles? Maybe choose something else more academically precise that does not require quotes eg "want" and "super mother".

Introduction, line 77: "What are the magnitude of brooding costs" maybe "What is the magnitude...?"

Results, sentence starting at line 280, "All the 26 egg sacs..." this sentence needs revision in more than one place.

Results, line 310 "...", however, under high level of sun exposure egg sac removed females seemed escaping in a little but not significantly slower speed than egg sac carrying females...". Please revise the sentence.

Results, line 313 maybe add a comma before "egg sac carrying mothers" and also add a hyphen "-" between egg-sac-carrying females? I suspect that in many instances in the manuscript, hyphenating "egg-sac" and "egg-sac-carrying" will help understanding and flow.

Discussion, line 371 to 373 several grammar aspects in different sentences.

Response: We thank the reviewer for their careful reading and have implemented all suggested language edits throughout the manuscript.

Changed "embryo development" and "embryo cues" to "embryonic development" and "embryonic cues" (Lines 64, 77).

We have removed the anthropomorphic language and rephrased the sentence to: "Thus, it is reasonable to posit that eggs benefit from a brooding mother who can respond to their developmental needs." (Line 66)

Corrected to "What is the magnitude of brooding costs..." (Note: This line was part of a broader revision of the aims/hypotheses, and the specific phrase has been reworked for clarity).

The sentence starting at line 289 has been rewritten for clarity: "All the 26 egg-sacs brooded by biological mothers and all twelve egg-sacs brooded by conspecific foster mothers were opened on time (Fisher's exact test: $P = 1$), and their eggs showed similar hatching probabilities..." (Lines 289-292)

The sentence on line 320 has been revised for clarity and accuracy: "under high level of sun exposure, egg-sac-removed females escaping speed did not significantly different with egg-sac-carrying females..." (Lines 320-321)

We have added hyphens for compound adjectives as suggested (e.g., egg-sac-carrying females). This has been done consistently across the entire manuscript.

The sentences in the paragraph from line 371 have been carefully revised to correct grammar and improve flow (Lines 380-382).

In addition to the points highlighted by the reviewer, we have performed a thorough copy-edit of the entire manuscript to correct grammar, improve sentence structure, and ensure consistency in terminology.

Second decision letter

MS ID#: bio.062232R1

MS Title: Egg-sac-brooding wolf spiders show flexible hatchling emergence and context-dependent escape performance

Authors: Bai-Lu Chen, Jing-Xin Liu and Zhanqi Chen

© 2025. Published by The Company of Biologists under the terms of the Creative Commons Attribution License (<https://creativecommons.org/licenses/by/4.0/>).

I have now reached a decision on the above manuscript.

The reviewer reports are shown at the bottom of this email or can be accessed, together with a copy of this decision letter, by going to:

Both reviewers appreciated the substantial improvements you made to the manuscript. However, they pointed out some very minor typographical errors that should be corrected before I accept the manuscript.

At this stage, we also ask you to ensure your manuscript complies with our formatting guidelines “ please see our manuscript preparation guidelines for details. Provided you are able to fully address the referees’ comments, we are positive about publication of your paper (we accept over 95% of revision submissions) and therefore hope you won’t mind any extra work involved in reformatting your manuscript at this point.

Please upload both a 'clean' version of your Word file, along with a highlighted version clearly showing where you have made changes in the revised manuscript. Please avoid using 'Track changes' in Word files as these are lost in PDF conversion.

I should be grateful if you would also provide a point-by-point response detailing how you have dealt with the points raised by the reviewers in the 'Response to Reviewers' box. Please attend to all of the reviewers’ comments. If you do not agree with any of their criticisms or suggestions please explain clearly why this is so.

Reviewer 1

Comments for the author

Thank you for thoroughly addressing my comments, including the raw data, and editing the manuscript. I'm satisfied with all the changes. I only have a few structure/grammatical comments:

Lines 20-22: I believe grammatically you don't require question marks on these three statements. For the Summary, there is a word "and" missing and in case you are already at word limit, you could potentially do this:

"Wolf spider mothers flexibly open egg-sacs in response to embryonic cues and modulate escape speed under acute risks, illustrating a context-dependent trade-off during parental care.

Line 320: "did not significantly differ" (not "different")

Lastly, the figures letter order is inconsistent between figure 5 and 6. In case the journal has requirements for this, I've noticed:

In figure 5, the letter sequence goes from left to right

AB

CD

EF

Whereas figure 6 letter sequence goes from up to down

ACE

BDF

Reviewer 2

Comments for the author

The authors addressed all of my comments and the paper has improved.

Minor comments:

In the Abstract: please verify if the question marks are necessary.

For clarity and to facilitate data interpretation, please briefly explain the statistics in the figures legends of Fig. 3 and Fig. 6. (specifically: briefly explain what the letters "a, b, c" represent and which statistical tests were applied). Please add the N of each experiment somewhere in the figures themselves (eg, at the bottom of each group), if possible.

In Fig. 3 and its figure legend, I am not sure you need the apostrophe (and the "s") in "eggs" or "egg's". For instance, "Effects on egg-hatching rate" reads OK.

Reviewer's Responses to Questions

Experimental quality

Does each figure have the proper controls?

If 'No', please indicate reasons in Comments for Author box below.

Reviewer #1:

- Yes

Reviewer #2:

- Yes

Were the data analyzed using appropriate statistical tests?

If 'No', please indicate reasons in Comments for Author box below.

Reviewer #1:

- Yes

Reviewer #2:

- Yes

Reproducibility

Were experiments performed using adequate number of biological replicates?

If 'No', please indicate reasons in Comments for Author box below.

Reviewer #1:

- Yes

Reviewer #2:

- Yes

Does the methods section provide sufficient detail to permit reproducibility?

If 'No', please indicate reasons in Comments for Author box below.

Reviewer #1:

- Yes

Reviewer #2:

- Yes

Completeness

Are the manuscript's conclusions supported by the data?

If 'No', please indicate reasons in Comments for Author box below.

Reviewer #1:

- Yes

Reviewer #2:

- Yes

Scholarship

Do the authors cite and discuss the merits of data that would argue for and against their conclusion?

If 'No', please indicate reasons in Comments for Author box below.

Reviewer #1:

- Yes

Reviewer #2:

- Yes

Does the manuscript title & abstract accurately reflect the contents of the manuscript, without hyperbole?

If 'No', please indicate reasons in Comments for Author box below.

Reviewer #1:

- Yes

Reviewer #2:

- Yes

Second revision

Author response to reviewers' comments

Reviewer 1: Thank you for thoroughly addressing my comments, including the raw data, and editing the manuscript. I'm satisfied with all the changes. I only have a few structure/grammatical comments:

We thank the reviewer for the careful reading and constructive suggestions. We have implemented the following changes:

Lines 20-22: I believe grammatically you don't require question marks on these three statements.

Response: Modification done.

For the Summary, there is a word "and" missing and in case you are already at word limit, you could potentially do this:

"Wolf spider mothers flexibly open egg-sacs in response to embryonic cues and modulate escape speed under acute risks, illustrating a context-dependent trade-off during parental care.

Response: Modification done.

Line 320: "did not significantly differ" (not "different")

Response: Modification done.

Lastly, the figures letter order is inconsistent between figure 5 and 6. In case the journal has requirements for this, I've noticed:

In figure 5, the letter sequence goes from left to right

AB
CD
EF

Whereas figure 6 letter sequence goes from up to down

ACE
BDF

Response: We thank the reviewer for pointing out the inconsistency in the subplot labeling order between Figure 5 and Figure 6.

In Figure 5, the subplots are arranged in a left-to-right, top-to-bottom sequence (A-B, C-D, E-F) to clearly illustrate the developmental progression and outcomes of egg-sacs under different brooding conditions.

In Figure 6, the subplots are arranged in a column-wise sequence (A-C-E, B-D-F) to facilitate direct comparison of posterior distributions (left column) and conditional effects (right column) across the three risk scenarios (sun exposure, flood, predator).

This arrangement was intentional to enhance visual clarity and interpretability. However, we will ensure that the final version conforms to the journal's specific formatting guidelines if required.

Reviewer 2: The authors addressed all of my comments and the paper has improved.

We thank the reviewer for the careful reading and constructive suggestions. We have implemented the following changes:

Minor comments:

In the Abstract: please verify if the question marks are necessary.

Response: We have removed the question marks from lines 20-22 and rephrased the statements as declarative sentences for better grammatical flow.

For clarity and to facilitate data interpretation, please briefly explain the statistics in the figures legends of Fig. 3 and Fig. 6. (specifically: briefly explain what the letters "a, b, c" represent and which statistical tests were applied). Please add the N of each experiment somewhere in the figures themselves (eg, at the bottom of each group), if possible.

Response: We have updated the figure captions as follows:

For Fig. 3: Statistical note: Letters (a, b) above violins indicate credible groupings based on Bayesian logistic mixed model (BLMM) posterior distributions. Groups sharing the same letter do not differ credibly (95% CI-HPD includes zero). Sample sizes (N) for each group are indicated below the group labels.

For Fig. 6: Statistical note: Bayesian linear models (BLMs) were used to compare escape speeds. Letters (a, b) indicate credible groupings. Sample sizes (N) for each group are provided in the figure.

In Fig. 3 and its figure legend, I am not sure you need the apostrophe (and the "s") in "eggs" or "egg's". For instance, "Effects on egg-hatching rate" reads OK.

Response: We have replaced "eggs' hatching rate" with "egg hatching rate" throughout Figure 3 and its caption for clarity and grammatical correctness.

Third decision letter

MS ID#: bio.062232R2

MS Title: Egg-sac-brooding wolf spiders show flexible hatchling emergence and context-dependent escape performance

Authors: Bai-Lu Chen, Jing-Xin Liu and Zhanqi Chen

I am happy to tell you that your manuscript has been accepted for publication in Biology Open, pending our standard publication integrity checks. It was accepted on 7th October 2025.